# Reflections on Experiencing Parental Bereavement as a Young Person: A Retrospective Qualitative Study

**DOI:** 10.3390/ijerph19042083

**Published:** 2022-02-13

**Authors:** Angel Marie Chater, Neil Howlett, Gillian W. Shorter, Julia K. Zakrzewski-Fruer, Jane Williams

**Affiliations:** 1Institute for Sport and Physical Activity Research, Centre for Health, Wellbeing and Behaviour Change, University of Bedfordshire, Bedford MK41 9EA, UK; julia.fruer@beds.ac.uk (J.K.Z.-F.); jane.williams@beds.ac.uk (J.W.); 2Centre for Behavioural Medicine, University College London School of Pharmacy, London WC1H 9JP, UK; 3Department of Psychology, Sport and Geography, School of Life and Medical Sciences, University of Hertfordshire, Hatfield AL10 9AB, UK; n.howlett@herts.ac.uk; 4Centre for Improving Health Related Quality of Life, School of Psychology, Queen’s University Belfast, Belfast BT7 1NN, UK; g.shorter@qub.ac.uk

**Keywords:** bereavement, death, parent, young person, grief, emotion, post-traumatic growth

## Abstract

**Background**: It is estimated that approximately 41,000 children and young people experience the death of a parent each year. Grief responses, such as anxiety and depression, can follow. This research investigated the adult reflections of experiencing parental death as a young person. **Methods**: Semi-structured interviews were conducted with adults (*N* = 14; female *n* = 8) who experienced parental death as a young person, which occurred over 5 years ago (time since death, *M* = 12.9 years; age at death, *M* = 16.4 years; age at interview, *M* = 30.9 years). The data were analysed inductively using thematic analysis. **Results**: Seven themes revealed that parental bereavement can lead to (1) “*Distance and isolation*” and is an (2) “*Emotional journey*” with (3) a “*Physical impact*”. Many experienced (4) “*Post-traumatic growth*” but acknowledged that (5) “*Life will never be the same*”, highlighting the importance of (6) “*Support and understanding*” and triggers for (7) “*Re-grief*”. **Conclusions**: Parental bereavement has significant emotional and physical consequences, but can also lead to personal growth. Talking therapies were rarely accessed, often due to a lack of awareness or desire to engage, revealing a translational gap between existing support services and uptake. Enabling open conversations about grief and identifying suitable support is a public health priority. This need has been amplified since the start of the COVID-19 pandemic, which may be a trigger for grief empathy and re-grief in those who have already been bereaved.

## 1. Background

Before the COVID-19 pandemic, there were 616,014 registered deaths during 2018 in the UK alone [1,2,3]. In the most recent available national statistics [4], it is estimated that 23,600 parents die annually, leaving behind dependent children. This equates to approximately 41,000 children and young people who experience the death of a parent each year [4]. Every 22 min within the UK, a child or young person will experience the death of a parent and 1 in 20 young people have experienced parental or sibling bereavement by the age of 16 years [5]. This is likely to have increased since the start of the COVID-19 pandemic, however, official statistics are not routinely collected and the disclosure of dependents is not a current requirement at the time of registering a death in England. Furthermore, bereavement has not been routinely measured as an adverse childhood experience (ACE) when investigating national trends [6,7] and the subsequent impact on physical and psychological health. This may impact on reports of the number of those who have been bereaved and referred to support services.

The parental figure(s) are integral to holding families together [8] and their death can create uncertainty and devastation. After parental bereavement, individuals may experience one or many grief outcomes [9], such as anxiety, depression, guilt, loneliness, isolation, suicidal ideation, substance use, insomnia, aggression, post-traumatic stress, lowered self-esteem, decreased well-being or resilience, emotional eating, self-harm, lack of concentration and a lower quality of life [8,10,11,12,13]. These experiences could also differ, depending on the type of loss [5,8]. Bereavement is unique to each person and there is no known single defining experience or duration. Interventions can, however, offer benefits to those who have been bereaved at a young age, with evidence from a systematic review [14] showing improvements in complicated grief and several grief outcomes through interventions with ages ranging from 0–18 years.

Young people often need assistance in adjusting to a new life without their parental figure [15], which may include relocation (e.g., moving home/school) and peer support [16,17]. The loss of a loved one early in life can lead to further re-grief once they have adjusted to the death [18,19]. Events such as graduating from university, getting married, buying a house or having children could trigger this response [8], whereby the individual is unable to share a meaningful event with their person and experiences of grief re-surface. Exposure to wide-scale death, such as the reporting of daily COVID-19 fatalities, and family loss could also trigger grief empathy and re-grief responses [20].

Those who have been bereaved should receive suitable support to help to manage their grief outcomes. There are hundreds of bereavement support services available in the UK alone, with most organisations offering talking therapy (i.e., individual counselling, cognitive behavioural therapy [CBT] or group counselling) or family support [21]. Residential “grief camps” are also available and have been found to support young people aged 9–25 years who have experienced the death of a parent within the previous 2–15 years [10], with only four participants who had experienced the loss of a parent less than five years previous. This opens up questions as to when the ‘right’ time would be for such support. This service, and others like it, are offered to a wide range of ages [14] and time since loss. It is, therefore, important to understand parental bereavement across the full age range of a young person to look for commonalities of experience in order to extend support service provision. The World Health Organisation (WHO) defines a young person as being between the ages of 10–24 years [22].

While details for bereavement services are readily available, research [21] has confirmed that it is unclear how often they are accessed, by how many, whether they are acceptable to young people and what support would be welcomed from those who have experienced the loss of a parent. This study aimed to understand the experience of parental bereavement, what support was received and what should be considered, with a view to inform future interventions and service development. This study sought to answer the following research questions: (1) What was it like to experience the death of a parent as a young person? (2) What support was accessed and received for parental bereavement? Given the uniqueness of bereavement and allowing time for adjustment, this study focused on adult reflections from at least five years after the parental death.

## 2. Method

### 2.1. Design

A qualitative approach using semi-structured interviews was used to enable a deep exploration into the experience of parental bereavement. The interviewer (J.W.: female, experienced in qualitative research) had not experienced the death of a parent and, therefore, did not bring the potential bias of past experience into the interviewing or analysis process. Two members (A.M.C., G.W.S.: both female) of the research team had been parentally bereaved, one of whom was at the age under investigation (A.M.C.). A.M.C. has also supported her two children through parental bereavement following the sudden death of their Dad (at ages 13 and 20 years). N.H. (male) and J.K.Z-F. (female) had not experienced parental death and, therefore, acted as bracketers (a bracketer alleviates any preconceptions arising from direct experience that may influence the research process [23]). This study is reported using the 32-item checklist of the consolidated criteria for reporting qualitative research (COREQ-32) [24]. University ethical approval was provided in December 2017 (reference number: 2017ISPAR008).

### 2.2. Participants

Taking past research [10] into consideration, the current study focused on an adult population (over 18 years old) who had experienced the death of a parent or guardian as a young person between the ages of 10–24 years (using the WHO definition [22]). Although there is no time limit given to the grieving process, this study aimed to recruit those who had had at least five years to adjust to the death of their parent.

Participants were recruited using a convenience sample via social media posts (e.g., Facebook, Twitter) and a snowballing strategy. A total of 14 participants (eight females; six males) participated in the interviews, which took place from March 2018 to September 2018. Participants were not known to the interviewer and, at the time of the interview, were between the ages of 21–41 years old (*M* = 38.62 years; *SD* = 6.69) with a mean age at bereavement of 16.42 years old (*SD* = 4.17). This includes the details of one participant who asked for their details not to be shared in the participant characteristics table but consented to the detail being used as a pooled descriptive. Table 1 provides further participant details, including the type of bereavement, age at bereavement and pseudonyms. Most participants were British (*n* = 12) with the remaining two being from other ethnicities. Religious beliefs held were: Christian (*n* = 3), Protestant (*n* = 1), and Catholic (*n* = 1). The remaining were not religious (*n* = 7). Participants experienced both expected (*n* = 5) and unexpected (*n* = 9) parental death. The majority (*n* = 12) experienced parental bereavement from illness, with one unknown and one natural cause. There was an equal number of the loss of maternal (*n* = 7) and paternal (*n* = 7) figures. Based on the preferences of the participants and the terms used in their narratives, parental figures will be termed Mum and Dad.

### 2.3. Materials and Procedure

Following contact and informed consent, the participants were asked, via a short survey, to provide information on who died, how and when they died and if the death was expected. Tick boxes were used to minimise the cognitive effort needed to process this information. These details were asked immediately prior to the interview so that the interviewer could begin with information about the loss without this being the first part of the interview. There were no changes to the interview schedule based on these responses. A semi-structured interview schedule was used, asking questions such as “*How did losing your [Mum/Dad] affect you?*”, “*What support did you receive after the loss of your [Mum/Dad]?*”, “*How did you come about receiving that support?*” The interview schedule was developed (by J.W. and A.C.) following informal PPI (patient and participant involvement) conversations with members of an advisory group for the wider programme of research [21] and participants from a residential bereavement support group where J.W. was a volunteer. Two digital audio recorders that were set to record at 44.1 kHz (Tescam Dr-05) were used to record the interviews, which lasted between 24 min and 63 min (*M* = 44.04; *SD* = 12.82) and were then transcribed verbatim following each interview. Data saturation was reached by the 11th interview, whereby no new themes were identified from the following three interviews. At this point, the sample size was agreed upon, reflecting recommendations from the literature [25], and data collection ceased.

Due to the sensitive nature of the interviews, the environments (university campus (*n* = 6); phone call (*n* = 3); participant’s home (*n* = 2); local café (*n* = 2); participant’s work (*n* = 1)) were set up to minimise negative affect, with access to fresh air, tissues and water. The researcher was mindful not to wear dark (mourning-style) outfits and of the participants’ emotional state, monitoring for signs of distress. A protocol was developed to respond appropriately: to stop the interview if the researcher felt that the participant was in distress and, following the interview, to give the participants a full verbal debrief and a list of bereavement support services. Additionally, participants received a follow-up phone call the day after the interview as a further “check in” phase of the debrief. Finally, the researcher, a psychology postgraduate, engaged in regular supervision, including a phone call or meeting following each interview, from a HCPC registered practitioner psychologist (A.M.C.) to ensure her own mental health needs were being met.

Assurance was given that all personal information provided by the participants would be kept confidential, with interview transcripts being anonymised and pseudonyms given. The British Psychological Society’s [26] Code of Ethics and Conduct was followed throughout. Personal information, such as phone numbers and GP details, were recorded and participants were informed that this information would only be used if the interviewer felt there was a need to break confidentiality (i.e., if they believed there was a risk of harm to self or others). There were no incidences where confidentiality needed to be broken.

### 2.4. Analysis

The interviews were transcribed verbatim, coded (by J.W.) and double coded (by A.M.C.) using the qualitative data analysis software NVivo [27]. An inductive approach was used within the interview transcripts to condense the data into nodes, developing clear links to the research question, followed by a thematic analysis [28] to analyse the transcripts, identifying central themes. A thematic map (Figure 1) was developed by J.W. and independently checked against the full NVivo data by A.M.C. It was amended three times through an iterative process to reach the final agreed themes.

## 3. Results

Seven themes portrayed the journey experienced following the death of a parent: (1) “Distance and isolation”; (2) “Emotional journey”; (3) “Physical impact”; (4) “Post-traumatic growth”; (5) “Life will never be the same”; (6) “Support and understanding”; and (7) “Re-grief”. Underneath are quotes that support the themes in parentheses, along with participant pseudonyms, which parent died and the age of the participant at the time of death. A visual representation of the themes can be seen in Figure 1.

### 3.1. Theme 1: Distance and Isolation

#### 3.1.1. People Don’t Know How to React

Many participants noted that following the parental death, their peers often did not know how to react or behave around them. They stated that their friends would be very tentative around the topic of their parent and not know what to say. Although many stated their friends did this without the intent to cause harm, their lack of knowing how to react was painful.


*“I think they just didn’t know what to say, so they didn’t…”*
(Gail, 24 years old when her dad died)


*“I was quite young, so I think a lot of my friends didn’t know how to act…I think you just kind of like pussyfoot around it…”*
(Laura, 11 years old when mum died)


*“I think it is that they don’t know what to say and it is just really difficult, and it almost becomes like they are stepping on eggshells around me.”*
(Kate, 18 years old when her mum died)


*“So, people who have never lost a parent, they feel so sorry when I mention it and they get really panicky and they are like “OMG, what do I do, what do I do.”*
(Louise, 13 years old when her dad died)


*“They have this look, where they feel sorry for you, and that pisses me off. Because they don’t get it.”*
(Marie, 20 years old when her dad died)

Whilst peers in their youth may not have been the best support, those who had experienced the death of a parent were able to understand emotions and what they were going through.


*“They [people who have not been bereaved] say “I’m sure in a couple of days you’ll feel better”. No, actually, I won’t feel better, no, that’s crap don’t talk to me… People who are in that situation understand. It is like an unknowing acknowledgment that the person will always be part of your life… people who have never been in that situation, they are very naïve to it.”*
(Marie, 20 years old when her dad died)

#### 3.1.2. Distancing from Others

With the view that peers who had not experienced a parental death at their time of loss did not understand or know how to respond, participants noted that they would distance and isolate themselves, finding it easier to not have the conversations.


*“I isolated myself and didn’t want to hang out with people as much as I used to… I was just really quiet, although I had friends, I would prefer to be by myself.”*
(Louise, 13 years old when her dad died)


*“I didn’t wanna go back [to work], I didn’t want to face anyone.”*
(Claire, 22 years old when her mum died)


*“So, it is easier to just not speak to people. Because genuinely you are not feeling alright. You hate your life and miss the people you have lost. You wanna start again and technically the only way to start again with those people is not to be in the world and you do genuinely have those thoughts like over and over again…”*
(Marie, 20 years old when her dad died)


*“I just shut down. I couldn’t go anywhere. I couldn’t leave the house. I was just paralysed, I couldn’t do anything.”*
(Ben, 18 years old when his dad died)

#### 3.1.3. Blocking It Out

There was a common feeling of not wanting the death to be at the forefront of their mind. To cope, they blocked out their emotions by distracting themselves or giving themselves a new focus.


*“I’m the kind of person who will keep myself busy to avoid feeling about it sometimes and I think that was kind of how I coped with it. So, my mental health wasn’t as great as it could have been at the time.”*
(Kate, 18 years old when her mum died)


*“I did a lot of travelling. My sisters would say that was because I was struggling and I needed to be away.”*
(Zara, 23 years old when her mum died)


*“I wasn’t letting any of my emotions through. So, they were just building up, building up and building up and all of a sudden I can’t get out of bed… To be honest, really I have crammed any single minute in my life to work. So, I don’t have to think about all my emotions. So, I just work constantly.”*
(Marie, 20 years old when her dad died)


*“How I didn’t cry and how I didn’t lose myself, I just don’t know. I just felt like he wouldn’t want me to feel bad that he is gone…”*
(Jack, 14 years old when his dad died)

### 3.2. Theme 2: Emotional Journey

Each participant experienced unique emotions after the death of their parent, with no two having the same emotional journey yet all acknowledging that it was emotional. Denial and disbelief, anger, anxiety, depression, guilt and devastation were among the emotions that were experienced.

#### 3.2.1. Denial

After experiencing the death of their parent, participants felt that they detached from the experience as if it were not real. Many believed it was an experience that only happened to other people and that losing a central figure, such as a parent, would not happen to them.


*“My mum was in the chapel of rest and I was trying to wake her up and I am just screaming this can’t be real.”*
(Claire, 22 years old when her mum died)


*“It was like everything was in slow motion, just it is almost surreal. It is just, yeah, you are almost looking at yourself outside you know, it’s like, I don’t know, you are almost part of a film.”*
(Ben, 18 years old when his dad died)

#### 3.2.2. Anger

Anger was evoked and needed an outlet, which turned into self-destructive behaviour at times.


*“I was an angry 20-year-old. Angry. Angry. Angry… I think I had a lot of anger that I didn’t have anywhere to put. I had nowhere to vent it out, so I was taking it out on myself. Self-harming reduced that anger a hell of a lot.”*
(Marie, 20 years old when her dad died)


*“I was really sort of frustrated. Really angry I went through a really sort of angry, angry phase.”*
(Jim, 17 years old when his mum died)

#### 3.2.3. Anxiety

There was a shared anxiety about dying, something that was not felt before the death of their parent.


*“I am completely paranoid that I am going to die.”*
(Claire, 22 years old when her mum died)


*“I was never anxious or I was like a very upbeat person, but somehow after that I became more anxious.”*
(Louise, 13 years old when her dad died)


*“I suffered with anxiety, quite badly.”*
(Marie, 20 years old when her dad died)

#### 3.2.4. Depression

A sadness was experienced, including both general sadness and clinical depression.


*“I wouldn’t say I was depressed, but obviously there are days where I have felt a bit lonely and I have thought this would be so different if my mum was here.”*
(Laura, 11 years old when her mum died)


*“Grieving can be quite a deteriorating process it can be quite a decaying, depressive period and once that depression gets hold of you. I think the worse…”*
(Greg, undisclosed age when his dad died)


*“I was diagnosed with depression [following parental death]…”*
(Ben, 18 years old when his dad died)


*“It did sink me into a depression, I was on anti–depressants”*
(Gail, 24 years old when her dad died)

#### 3.2.5. Guilt

Guilt was another common feeling shared by the participants. Some individuals felt guilty about living their lives and creating happy memories when they felt they should be sad and upset about the death of their parent, whereas others felt guilty for the things they did or did not say to their parents before they died. Sharing true feelings with people also created a sense of guilt.


*“I felt really guilty because I had only just moved out the November before.”*
(Gail, 24 years old when her dad died)


*“The guilt at things you said, things you didn’t say, you attempt to make amends for”*
(Greg, undisclosed age when his dad died)


*“I always remember feeling really guilty about it in the really early stages about enjoying myself. Having fun, I felt really guilty about it.”*
(Jim, 17 years old when his mum died)


*“Then I feel guilty for her [a friend] cos she is probably thinking “Shit, the poor woman’s going to commit suicide. I better tell her she is loved.””*
(Marie, 20 years old when her dad died)

#### 3.2.6. Devastation

Participants explained that the death of their parent turned their world upside down and that they could not imagine anything worse.


*“How did it affect me; it literally broke me. Broke my whole entire world.”*
(Marie, 20 years since her dad died)


*“From that day, everything just got flipped on its head.”*
(Claire, 22 years old when her mum died)


*“I feel like that [the death of a parent] is probably the worst thing that could ever happen to anyone…”*
(Laura, 11 years old when her mum died)

### 3.3. Theme 3: Physical Impact

The impact of parental bereavement was not just described as emotional. There were shared behavioural outcomes of grief, such as changes in eating habits, drug use and sleeping problems. Physical illness was also felt to be a physical response that was linked to the grieving process.

#### 3.3.1. Maladaptive Behaviour


*“I just couldn’t be bothered, I wasn’t, I just did not get hungry for ages and I struggle with food now if I get stressed. I just don’t eat, just don’t get hungry.”*
(Claire, 22 years old when her mum died)


*“I don’t think I ate for a long time. So, I lost a lot of weight. I was only little anyways.”*
(Marie, 20 years old when her dad died)


*“It would be right in front of the telly, eat whatever the hell I want. Whatever, whenever as much as I wanted. I didn’t care, so yeah I probably put on a stone, maybe two.”*
(Gail, 24 years old when her dad died)


*“Marijuana worked really well for the first 10 years maybe in keeping my emotions, and then it started not to work…You couldn’t make it to the bed or shower, or you couldn’t feed yourself.”*
(Greg, undisclosed age when his dad died)


*“Mostly be exhausted and you would be drained because you were always, if you were not crying you were always talking to people…”*
(Rebekah, 16 years old when her mum died)

#### 3.3.2. Physical Reactions


*“Loads of panic attacks in places like if I was stuck in a lift, if I was in a lift and the lift was taking too long or if I was in a lecture theatre sandwiched between people. I would start to panic and stuff like that…”*
(Zara, 23 years old when her mum died)


*“Then I got diagnosed with Fibromyositis Obviously, there is not much research into it but they said one factor… most people get diagnosed with it when there has been a big trauma or loss in their life”*
(Claire, 22 years old when her mum died)

### 3.4. Theme 4: Post-Traumatic Growth

With the interviews occurring over five years after the bereavement, participants reported that they were stronger and more resilient when faced with hardships. In a sense, the trauma from their loss had led to them “growing” as a person. It was also noted that the influence their parents had on them never disappeared and remained with them.

#### 3.4.1. Life Perspective

Individuals often discussed having a new outlook on life. This included having a different perspective on life and how they should be living it, changes to what pathways they should take and their views about the world. Some felt uncertainty of what life would hold, particularly those who experienced an unexpected or sudden bereavement.


*“Life is for living and it is for living now because you never know what is going to happen.”*
(Zara, 23 years old when her mum died)


*“I try and sort of like take opportunities more. I do new things all the time. I try and sort of make the most of everything.”*
(Jim, 17 years old when his mum died)


*“You don’t know what is around the corner. Like everything can change in the blink of an eye.”*
(Rebekah, 16 years old when her mum died)


*“I would be a much more selfish person now if I hadn’t gone through all that. I was a spoiled little brat as a kid. I was a Daddy’s girl and would snap my fingers and get what I wanted. Now I realise you can have stuff taken away from you really easily, so I appreciate it more.”*
(Marie, 20 years old when her dad died)

#### 3.4.2. Builds Resilience

With regards to the long-term adjustment to a life without their parent, participants felt that they had become stronger, although this was not something about which they felt they had a choice. They also had less sympathy for people who are unable to overcome what they saw as little challenges.


*“I can stop any sort of nonsense in my life. Any maybe pitfalls that other people might have fallen into. You sort of harden up… I felt like I had to be sort of strong and just cope on my own. Sort of and become more self-sufficient almost.”*
(Ben, 18 years old when his dad died)


*“Think I am so much stronger than what I would have been…”*
(Laura, 11 years old when her mum died)


*“I have a high level of resilience cos I have had to. I have had no choice. I have had to overcome things.”*
(Marie, 20 years old when her dad died)


*“I had to start learning to speak up for myself, cos I was very shy, so it was like well no one else is going to talk up for me. I better do it for myself.”*
(Gail, 24 years old when her dad died)

#### 3.4.3. Relationship Strength

Many reported that their families had grown in strength and became closer to each other, creating a tighter family bond. However, some noted that their family relationship had deteriorated and that family members had drifted apart.


*“I say certainly the bereavement and the loss of my Dad definitely made us a lot closer.”*
(Ben, 18 years old when his dad died)


*“We have got much closer because of it. Obviously after Dad we got closer but then after mum went, we got much closer cos we went, ‘it is just the two of us now’.”*
(Gail, 24 years old when her dad died)


*“I think one thing it did surprisingly, it brought my family together.”*
(Louise, 13 years old when her dad died)


*“Our family is disintegrated since my mum died.”*
(Zara, 23 years old when her mum died)


*“She left a massive hole in both families when she passed away.”*
(Adam, 13 years old when his mum died)


*“My family’s kind of quite diffused at this point.”*
(Chris, 21 years old when his dad died)


*“I think my Dad was obviously going through his own stuff and I was going through my stuff and we didn’t gel at the time.”*
(Kate, 18 years old when her mum died)

### 3.5. Theme 5: Life Will Never Be the Same

Whilst some individuals stated that they had grown and flourished from the trauma of the death of their parent, they also acknowledged that life would never be the same again. Individuals had to grow up quickly and often inherited more responsibilities. Despite this, the influence their parents had on their life endured.

#### 3.5.1. Growing Up Quickly

The need to look after other family members, such as the surviving parent or siblings, maintain the home and organise themselves in a way that would usually be the responsibility of an adult made participants mature quickly. There was a feeling that this was not something they were prepared for, but something that needed to be done.


*“I did feel I had to grow up quickly.”*
(Kate, 18 years old when her mum died)


*“I was looking after my little brother. Doing the things that my mum would have done for him… making sure the school trip has been paid for and stuff like that and I guess it left a bit of weight on my shoulders.”*
(Adam, 13 years old when his mum died)


*“You know just your mum and sisters need you, the house needs maintenance and the bills need to be paid and things like that need doing.”*
(Greg, undisclosed age when his dad died)


*“Well I was 22, and I wasn’t ready to grow up… and then I suddenly felt like I couldn’t leave my Dad.”*
(Claire, 22 years old when her mum died)


*“The main thing, trying to look after my sister… you realise all the things, she would have probably asked my mum.”*
(Rebekah, 16 years old when her mum died)


*“I felt like as now, as the head of the family, I had to pull myself together and get a job.”*
(Ben, 18 years old when his dad died)


*“Family wise, like I felt I had, you know, a great deal of obligation, that perhaps I hadn’t felt before that point.”*
(Chris, 21 years old when his dad died)

#### 3.5.2. Parental Influence Never Stops

The influence that the parents had had on their children was something that shaped and defined life choices even after their death. They often continued to pursue and follow the influence that their parents had set or highlighted that they missed the influence their parents would have had on their lives.


*“I think knowing he regretted certain things about his choices in life, made me think about mine a little more.”*
(Chris, 21 years old when his dad died)


*“I think that would have changed me taking a year out. I don’t think she would have let that happen. I think I would have been straight into doing University.”*
(Laura, 11 years old when her mum died)


*“She would have killed me if I decided to take a year out. She would have been like ‘no just go’.”*
(Kate, 18 years old when her mum died)

### 3.6. Theme 6: Support and Understanding

Social support was essential during the grieving process. Individuals looked to family, friends, other peers who had been bereaved and professional support, such as counselling and teacher support. Some turned to religion to help them through this period in their life and there was an overwhelming need to be loved.

#### 3.6.1. Feeling Loved and Supported

Individuals highlighted the need to feel wanted and supported during that time, especially after losing the unconditional love and support that their parent provided for them. This support came from family, friends, professional agencies and teachers. It was especially helpful from those who had shared a similar experience. However, despite the need for professional support, the majority of participants were not offered any or felt what they were offered was insufficient, and some felt that support drifted off as time went on.

#### 3.6.2. Family


*“I lent on Jason a lot. Jason my husband now, my boyfriend at the time. You know he supported me, cuddled me and helped me at the funerals, that sort of thing.”*
(Gail, 24 years old when her dad died)


*“I would go to him for support but it was actually the other way around I think in the initial stages. I think I dealt with it. Well not better, but I think my Dad just sort of, he was a bit more emotional to start off with, whereas I wasn’t.”*
(Jim, 17 years old when his mum died)

#### 3.6.3. Peers


*“We had a lot of friends who were very quick to sort of rally round my brother and I, to sort of be there for us.”*
(Adam, 13 years old when his mum died)


*“Friends-wise I think we all just got a lot closer, because everyone was just constantly making the effort to make sure we were okay and stuff.”*
(Rebekah, 16 years old when her mum died)


*“For the first couple of months, even the first year, everyone is so aware of what has happened… after a couple of months though you’re just left to like deal with it.”*
(Laura, 11 years old when her mum died)


*“I think the first week to me, like everyone around me rally’s round, they bring you dinner, not that you wanna eat, and they bring you flowers and cards, and you get daily phone calls from like the whole world and then the minute you have the funeral and everyone goes back to normal and you are just left, everyone just gets on with their life and you’re like, yeah your life is carrying on but this is just the beginning for me”*
(Claire, 22 years old when her mum died)

#### 3.6.4. Professional


*“They gave me like some leaflets and that was pretty much it as far as I remember. I think it was like sort of up to me if I wanted to get in touch they were always there. It was left open.”*
(Jim, 17 years old when his mum died)


*“I don’t think I was offered any counselling.”*
(Marie, 20 years old when her dad died)


*“I will always credit them [Physical Education/PE teachers] a lot for helping me through the rough time in various ways.”*
(Adam, 13 years old when his mum died)


*“She [PE teacher] just took more of an active role in making sure I was alright and sort of going above and beyond.”*
(Jim, 17 years old when his mum died)


*“Eventually I went back to school, you had the support of the guidance teacher.”*
(Rebekah, 16 years old when her mum died)


*“I didn’t really have any psychological support or help at all.”*
(Jack, 14 years old when his dad died)


*“They might have offered something, I’m 80% sure they didn’t.”*
(Greg, undisclosed age when his dad died)


*“They handed me a leaflet, but no one actually sat me down and said do you think this is something that you want to do. You know it was more handout a leaflet and that was it.”*
(Kate, 18 years old when her mum died)


*“Then we walked out of there, I had a booklet that was it. A booklet. I’m like, you know I have lost everything today and I have got a book.”*
(Claire, 22 years old when her mum died)

#### 3.6.5. Finding The Right Time to Talk

Whilst many of the participants were not offered or declined professional support, those who sought professional support did so at different times and for different reasons. Stigma and a lack of feeling able to engage were commonly cited as reasons for those who declined professional support.


*“I would wobble and was crying at like 3 o’clock in the morning, you know, what I need is to go and see someone about this cos no one here really understands.”*
(Kate, 18 years old when her mum died)


*“I don’t want talking therapy. I don’t want to talk about it all. I don’t feel comfortable talking about it. I think cos my family wasn’t like that, I just didn’t want counselling.”*
(Gail, 24 years old when her dad died)


*“I don’t think I would have taken it even if I was offered it. You know it is not until my partner recently died and it was a year and half after his death and he died 2 years ago I started to get counselling.”*
(Marie, 20 years old when her dad died)


*“I did, it was about a year afterwards.”*
(Ben, 18 years old when his dad died)


*“Just could not cope anymore, just could not cope.”*
(Greg, undisclosed age when his dad died)


*“Think I didn’t really feel like I needed it…”*
(Adam, 13 years old when his mum died)


*“So, seeing a therapist or kind of psychologist is the equivalent to being locked up in a mental institution, it’s just there is so much stigma about it. Even now in 2018”*
(Greg, undisclosed age when his dad died)


*“It would mean like it’s [seeking professional support] always on your record and it’s a lot of stigma around that”*
(Louise, 13 years old when her dad died)


*“I never went down that route because I just didn’t feel like I need to…”*
(Rebekah, 16 years old when her mum died)

#### 3.6.6. Religion

Many looked to religion to help to find support and understanding whilst others turned away from it, asking themselves why any greater power would cause such pain and suffering.


*“I didn’t care about anything. I was religious before my Dad died. Then hated God with a passion.”*
(Marie, 20 years old when her dad died)


*“But after this she [mum] started going to the mosque… but she wouldn’t do it religiously, she would do it in a spiritual way, and I think that helped her a lot.”*
(Louise, 13 years old when her dad died)

### 3.7. Theme 7: Re-Grief

Years after experiencing their bereavement, individuals had adjusted to their new life. However, there were many times when they began to experience grief outcomes again, such as feelings of anxiety, depression and a longing for their parent to be present. This re-grief often appeared at momentous events, perhaps at times when they had imagined that their parent would be there.

#### 3.7.1. Major Life Events

Participants wanted to be able to talk to and share the good and/or bad things that had happened in their lives with their parent, such as passing their driving test or having a baby.


*“It was weird, because I thought like daft things, like taking mum for a drive in a car and stuff when I passed, but obviously she wasn’t there to do that.”*
(Jim, 17 years old when his mum died)


*“I have missed my Mum and Dad more since I have become a parent, because I haven’t been able to introduce them to their grandchildren.”*
(Gail, 24 years old when her dad died)


*“Which then upsets me when I achieve something because I look around to tell someone and they are not there. Which I still find really hard.”*
(Marie, 20 years old when her dad died)


*“There isn’t anybody like your Mum… You can ask other people the same questions but there is no one who is going to be able to reply like your Mum.”*
(Zara, 23 years old when her mum died)

#### 3.7.2. Acceptance of a New Life

Participants noted that accepting their new lives without their parent was difficult, especially after particular life events, hearing a song or seeing a photo that triggered a memory. In many cases, this acceptance came with bitterness and they were filled with a sadness about their reality.


*“I am bitter. I am, I do get quite bitter, on my wedding day I just sat there crying. Then I went up there [the grave] the day after my wedding and took my bouquet up and put it on my Mum.”*
(Claire, 22 years old when her mum died)


*“All of a sudden, I just heard this song, very sad song, and I just started crying out of nowhere. Just bawling like a baby, you know and I had that feeling for like a day. I was like, bloody hell, where did that come from. I was like, is there anything else buried in there…”*
(Greg, undisclosed age when his dad died)

## 4. Discussion

Reflecting on the experience of parental bereavement, seven themes were identified in this research: (1) “Distance and isolation”; (2) “Emotional journey”; (3) “Physical impact”; (4) “Post-traumatic growth”; (5) “Life will never be the same”; (6) “Support and understanding”; and (7) “Re-grief”.

Distancing from others and blocking out emotions were seen as protective mechanisms against people who did not know how to react or behave towards grief. This is a common response [29], as is the anger that was expressed in the study. With no outlet, emotion was often turned inwards, resulting in maladaptive behaviour. Indeed, self-harm [13], suicidal ideation [30], substance use [10] and problems with eating [31] are often cited in the literature as grief outcomes, which is supported with evidence from the current study. Anxiety and a fear of dying were commonly felt by participants, alongside differing severities of sadness and depression. Anxiety and depression are also common to the grieving process [8] and are often a focus for intervention [32]. When individuals felt feelings of happiness, this led to feelings of guilt with the view that they should be constantly sad following their loss. Other research [33] also found this emotional guilt during positive experiences following the death of a parent and this is a potential area for future intervention to enable positive emotions and overcome feelings of guilt. This extends the literature that highlights mixed emotions following the death of a parent [34]. Kübler-Ross [35] proposed that there are five stages of grief (denial, anger, bargaining, depression and acceptance), echoing some of the emotions raised here. However, this model proposes unhelpful linearity and a lack of inclusion of wider aspects, such as guilt, re-grief and growth that are evidenced in this study.

Data from the current study can, in part, be conceptually likened to the notion of oscillating between loss orientation (confronting loss) and restoration-orientation (avoiding loss) from the Dual Process Model of grief [36]. Reflections from research, personal experience and the national grief seen since that start of the COVID-19 pandemic have led to the concept of “Your world and the ball of grief” [20] (see Figure 2). This analogy highlights re-grief and the impact that “grief empathy” (feeling for others who have experienced loss) could have on the re-grief process. At the time of a bereavement, it is considered that your world is almost completely consumed by a ball of grief, which can repeatedly hit a “pain” button, leading to grief outcomes. As time passes, rather than the ball of grief getting smaller, as has been postulated by others, this analogy suggests that the ball of grief stays the same but your world gets bigger, making it less likely that the ball will hit the pain button. During times of remembrance, the ball of grief may resurface and hit the pain button, causing re-grief. However, in this bigger world, that re-grief experience is less consuming because the ball of grief has more space to move around, thereby hitting the pain button less and causing less pain. Given the amount of loss experienced and publicised since the start of the COVID-19 pandemic, grief empathy could have been heightened during the pandemic. Evidence from the current study and the concept of re-grief due to triggers from grief empathy should be considered in future research.

The physical impact of grief was notable, particularly in relation to dietary changes. Food became both effortful and mindless, with respondents eating both less and much more than usual. This adds to the literature in other age groups, such as with elderly widows who see cooking as pointless and experience increased feelings of loneliness by eating alone [37], while others ate whatever they wanted, seeing life as too short not to do so.

The theme of post-traumatic growth (PTG) embraced the belief that the respondents could overcome anything, taking any opportunity presented to them and allowing them to step outside their comfort zones. This type of resilience has been described elsewhere [38], with PTG described as the positive changes individuals make after a traumatic event. This term is closely linked to positive psychology [39], where the main aim is to focus on building strengths, personal growth and flourishing. Future interventions could harness this concept of growth to enable positive coping methods following a parental death. This may, in turn, enable the acceptance of positive emotion following a bereavement.

Despite this growth and strength, it was acknowledged that life was not the same after the death of a parent. Some participants found themselves taking on additional responsibilities and having to grow up quickly. This is common [40], however, can add additional pressure, especially at an early age [41]. Despite the absence of their parent, it was clear that the influence they had had when they were alive still continued. This brought comfort to many, who shared that they would make decisions in their parent’s honour.

Finding the right time to talk and the right level of support was a debated topic. While there are many support services available [21], often the participants were either not offered support, they declined to take it up for fear of stigma, or that they would not be able to cope or they did not feel it necessary. Several mentioned being handed a leaflet or booklet, which they felt was not sufficient. Barriers to accessing bereavement services have been found elsewhere and include the availability of services, a lack of knowledge of services available, being unsure of how to access services, not feeling comfortable in asking for support and feelings of stigma [42]. Future research should investigate this translational gap and the interplay between the support received from professional services, peers and the school environment [43]. Such barriers to access support mirror those seen to talking therapies more generally [44,45] and, in particular, with those bereaved by sudden deaths such as suicide [46]. Some participants in this study received pastoral support via their schoolteachers or religious leaders; others were consoled by friends and relatives. Those who decided to use professional support services did so on their own terms when they felt ready or when they had reached a point where they needed it. This is something that should be encouraged from the onset and in wider society, with the encouragement of “open grief” [20] and the view that “*It’s ok not to be ok*”: a statement that has become more widely accepted in recent years, particularly within suicide prevention and mental health fields [47].

At times, the bereavement brought families closer together, while for others, it did not. There is a clear need for a consistent approach to be given to young people after the death of a parent that goes further than providing them with a leaflet or telephone number. Future public discourse and interventions must aim to reduce the perceived stigma attached to receiving support for grief and to bring people together in a supportive environment where there is a sense of belonging and understanding. It was clear from the interviews in this study that participants were most at ease when they were confiding in “someone like me”. While this offers support, feeling akin to someone else who has experienced a parental loss may also intensify feelings of grief empathy as the individual has insight into the complexity of grief, which may in turn trigger re-grief while recalling how they felt themselves. In a time of mass death, such as the COVID-19 pandemic, this warrants further investigation.

Bereavement support services are typically talking therapies (e.g., individual or group counselling) and evidence from this study suggests this approach is not always welcomed. An important consideration is whether more young people would seek bereavement support if there were an alternative approach. Music has been successfully used [48] to help support bereavement, showing this approach to provide a constructive outlet for emotions and to create connections. Physical activity has also been shown to provide support for grief outcomes [49]. Activities, such as those engaged in during residential weekends [50,51], running, martial arts [10] and extra-curricular activities [52], have been used to support young people who have experienced the death of a parent. Further, the use of online forums has shown promise, allowing young people to positively support each other following the death of a parent [53]. Expressive writing has also shown benefits for traumatic grief in adolescents who have been bereaved by war [54]. There is a need to examine whether alternative types of support could increase the engagement of young people and benefit the grieving process. These were not offered nor forthcoming avenues of support for the participants of the current study.

As the interviews were recorded over five years after bereavement, the participants described a point where they had begun to move forward with their grief. It was not a sense of ‘moving on’ and leaving the pain and memories of their parent in the past, but a sense of ‘moving forward’ and taking that pain and those memories with them along with a sense of control over the impact on their day-to-day lives. This supports the concept of oscillation in the Dual Process Model of grief [36]; however, they noted that major life events, such as passing a driving test, graduation, getting married or becoming a parent, would cause a sense of re-grief. This concept has been described before [8], highlighting a time when grief outcomes may be experienced after an individual has adjusted to the death of a loved one [18,19]. Techniques to cope with re-grief should form part of future intervention strategies.

By conducting semi-structured interviews, we enabled the participants to provide an enriched in-depth narrative of a sensitive topic, which could not be achieved with quantitative methods. As with other qualitative research, the experiences described here were not universal to all participants, nor would they be universal to all people who experience the death of a parent. However, data saturation was achieved as the experiences in each of the overarching themes were easily identifiable. It has been stated that, for qualitative research, ten plus three interviews is sufficient if saturation is achieved [25] and this research met that criterion. There is a dearth of previous research in the area of parental bereavement as a young person, the most comparable being that of Brewer and Sparks [50], who interviewed attendees at a grief bereavement organisation that Brewer had attended as a young person, capturing the experiences of both recently bereaved (*n =* 4) and those 10 years post bereavement (*n* = 9). While the age range of the current study sample could be critiqued due to the differences in developmental age and/or the time since the death, similar studies have drawn from such varied samples. Participants from a residential “grief camp” [50] who were aged between 9–25 years and had experienced parental death within 2–15 years were included in interviews and observations while attending the service together. The commissioning for children and young people’s (CYP) services is commonly combined and often, the interventions that are delivered for bereavement have a wide age range and access point regarding the time since the death [14]. It is, therefore, important to understand the experience of parental death from across all ages that may require access to such services in order to look for commonalities that these services should consider. While we appreciate that there is merit in comparing the experiences of the loss at similar ages, the approach and sample size would not allow this and would have moved the study away from the ultimate aim of producing commonalities across these ages to inform future intervention.

The current study extends the relatively limited literature base, offering the voices of those more recently bereaved than those in previous [51] research. However, in this sample, given the inclusion criterion of being at least 5 years post bereavement, an element of re-adjustment had occurred and there is an additional need to understand the experiences of those who have been bereaved within five years in order to support the development of future interventions. Future research should further aim to determine an optimal time for support following the death of a parent and ways to enable access to support services at a time that is right. This should be considered with an awareness of the nature of grief and a discussion on clinical concepts, such as “normal” and complicated grief.

## 5. Conclusions

Parental bereavement has significant emotional and physical consequences but can also support an individual to grow with purpose. There is no national method for collecting data on when a young person experiences parental loss and thus, there are no clear referral pathways to bereavement support services. This should be a national priority. Bereavement literature and talking therapies were rarely used by participants in this study, revealing a translational gap between existing support services and uptake that should be addressed. Alternative support services should also be offered and evaluated in more depth. This research provides evidence of the experience of parental bereavement and considerations for supporting such loss. These include enabling open conversations about death, offering support from others who have a shared experience and harnessing the benefits of outdoor space and physical activity. National and global grief are increasingly prominent in today’s society since the onset of the COVID-19 pandemic. Access to bereavement support deemed suitable by those who are in need of it should be seen as a public health priority.

## Figures and Tables

**Figure 1 ijerph-19-02083-f001:**
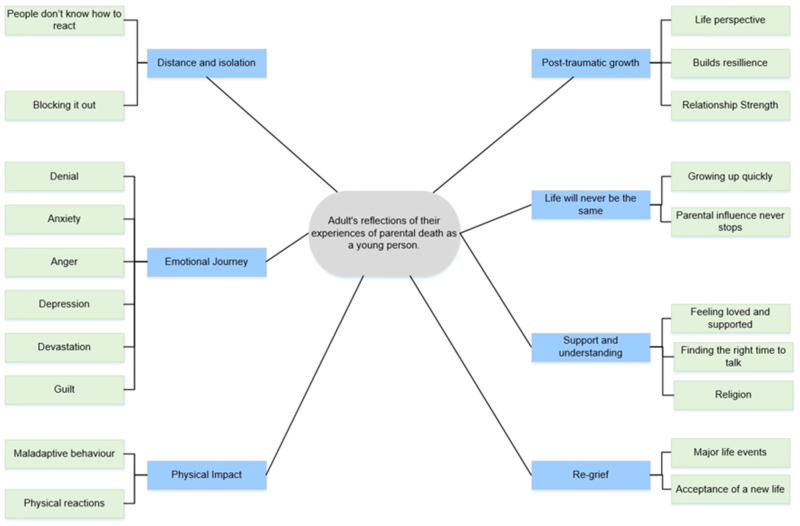
The thematic map investigating adult reflections on their experiences of parental death as a young person.

**Figure 2 ijerph-19-02083-f002:**
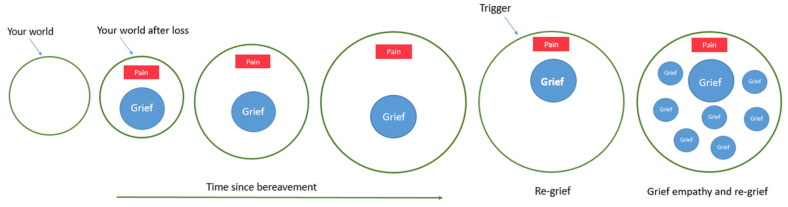
Your world and the ball of grief.

**Table 1 ijerph-19-02083-t001:** Participant characteristics, including their relationship to the deceased, details of the death and their age at the time of death.

Name	Age in Years (at Interview)	Parent Who Died	Cause of Death	Was Death Expected/Unexpected	Age in Years (at Death)	Years since Death
Marie	39	Dad	Illness	Unexpected	20	18
Ben	33	Dad	Natural	Unexpected	18	16
Rebekah	25	Mum	Illness	Unexpected	16	9
Laura	21	Mum	Illness	Unexpected	11	9
Jack	25	Dad	Unknown	Expected	14	12
Kate	25	Mum	Illness	Expected	18	8
Jim	28	Mum	Illness	Expected	17	11
Greg	Asked not to share	Dad	Illness	Unexpected	Asked not to share	Asked not to share
Zara	41	Mum	Illness	Expected	23	17
Adam	30	Mum	Illness	Unexpected	13	16
Chris	28	Dad	Illness	Expected	21	7
Claire	40	Mum	Illness	Unexpected	22	17
Louise	28	Dad	Illness	Unexpected	13	15
Gail	38	Dad	Illness	Unexpected	24	13

## Data Availability

The data presented in this study are available on request from the corresponding author. The data are not publicly available due to reasons of privacy and the sensitive nature of the research.

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
