# Peer review of "Reflections on Experiencing Parental Bereavement as a Young Person: A Retrospective Qualitative Study"

_ijerph, 2022, doi:10.3390/ijerph19042083_

Round 1

Reviewer 1 Report

Thank you very much for the opportunity to review this interesting manuscript. The paper aims to investigate experiences of parental bereavement in the young with special view on future intervention development and support services. The topic presents an important public health issue and the way it is addressed is novel. The work is based on a qualitative study referring to a data set of N=14 adult persons who experienced of parental death as a young person. It is an interesting, well-written paper with appropriate use of qualitative analyses. The article is following consolidated criteria for reporting qualitative research (COREQ-32). Semi-structured interviews were used to gather information about the experience of parental bereavement. Measures against a possible interviewer bias with regard to parental bereavement experiences were taken. Results are well structured. Identified main themes were: 1) “Distance and isolation”, 2) “Emotional journey”, 3) “Physical impact”, 4) “Post-Traumatic Growth”, 5) “Life will never be the same”, 6) “Support and understanding” and 7) “Re-grief”. Both, main themes and subthemes, are well presented in a way that is understandable for the reader. The authors discuss their results against the background of current literature. They conclude that parental bereavement has significant emotional and physical consequences, but can also support an individual to grow with purpose. Nevertheless, the manuscript could be improved in a number of areas. Please find attached my comments and suggestions.

Introduction:

  • The authors state that there is a lack of knowledge about how young people experience the loss of a parent and whether existing support services are sufficient. I would recommend the authors to go more into detail with regard to research findings in this area. For instance, what is known about the efficacy of support programs for children so far (e.g. Bergman A-S, Axberg, U, Hanson, E. BMC Palliat Care. When a parent dies - a systematic review of the effects of support programs for parentally bereaved children and their caregivers. 2017 Aug 10;16(1):39. doi: 10.1186/s12904-017-0223-y.)?
  • In my opinion, the reader would also benefit if more specific research questions were stated at the end of the introduction.

Methods:

  • Could you please give some more information about the development of the semi-structured interview guide (e.g. was it theoretical-based?)?

Discussion and Conclusions:

  • The authors state that their study suggests that bereavement support services such as talking therapies (e.g. individual or group counselling) would not always be welcomed. I agree with the authors that this could be one possible way of interpretation. Nevertheless, reading the results gave me the impression that there might have been a translational gap between the existence of bereavement support services and its actual use. I would recommend, to broaden up the discussion about this a bit more. Is there anything known (from your or from other studies) about possible barriers for support service uptake (e.g. lack of knowldege about bereavement support programs, anxiety of being stigmatized…). What makes it especially difficult for the young to take up support services? What could be done to your opinion to overcome a possible translational gap (e.g. involvement of teachers, family doctors)?
  • In line with the remark above I would strongly recommend to rephrase the sentence in the abstract “Talking therapies are seen as insufficient.“ From reading the quotes I believe that this is not exactly true, because study participants were not attending services and then evaluating them as being insufficient. It is my impression, that the quotes rather point to the fact that talking therapies were rarely attended revealing a translational gap between existing support services and uptake.
  • The authors point to the need of alternative approaches, such as music or physical activity. This is an important finding. Are there any data about the use of E-Mental health approaches for bereavement in this context (e.g. online self-help, or internet-based CBT)? Especially young people might be attracted to these approaches. 

Author Response

Reviewer: Thank you very much for the opportunity to review this interesting manuscript. The paper aims to investigate experiences of parental bereavement in the young with special view on future intervention development and support services. The topic presents an important public health issue and the way it is addressed is novel. The work is based on a qualitative study referring to a data set of N=14 adult persons who experienced of parental death as a young person. It is an interesting, well-written paper with appropriate use of qualitative analyses. The article is following consolidated criteria for reporting qualitative research (COREQ-32). Semi-structured interviews were used to gather information about the experience of parental bereavement. Measures against a possible interviewer bias with regard to parental bereavement experiences were taken. Results are well structured. Identified main themes were: 1) “Distance and isolation”, 2) “Emotional journey”, 3) “Physical impact”, 4) “Post-Traumatic Growth”, 5) “Life will never be the same”, 6) “Support and understanding” and 7) “Re-grief”. Both, main themes and subthemes, are well presented in a way that is understandable for the reader. The authors discuss their results against the background of current literature. They conclude that parental bereavement has significant emotional and physical consequences, but can also support an individual to grow with purpose.

Response: We thank the reviewer for their positive comments on our manuscript.

Reviewer: Nevertheless, the manuscript could be improved in a number of areas. Please find attached my comments and suggestions. Introduction: The authors state that there is a lack of knowledge about how young people experience the loss of a parent and whether existing support services are sufficient. I would recommend the authors to go more into detail with regard to research findings in this area. For instance, what is known about the efficacy of support programs for children so far (e.g. Bergman A-S, Axberg, U, Hanson, E. BMC Palliat Care. When a parent dies - a systematic review of the effects of support programs for parentally bereaved children and their caregivers. 2017 Aug 10;16(1):39. doi: 10.1186/s12904-017-0223-y.)?

Response: Thank you for asking us to expand on the points around efficacy of support programmes, which we have done.

  • Reviewer: In my opinion, the reader would also benefit if more specific research questions were stated at the end of the introduction.

Response: We have now added the specific research questions at the end of the introduction: What was it like to experience the death of a parent as a young person? 2) What support was accessed and received for parental bereavement?

Reviewer: Methods: Could you please give some more information about the development of the semi-structured interview guide (e.g. was it theoretical-based?)?

Response: The interview schedule was developed (by JW and AC) following informal PPI (patient and participant involvement) conversations with members of an advisory group for the wider programme of research, and participants at a residential bereavement support group where JW was a volunteer.

Reviewer: Discussion and Conclusions: The authors state that their study suggests that bereavement support services such as talking therapies (e.g. individual or group counselling) would not always be welcomed. I agree with the authors that this could be one possible way of interpretation. Nevertheless, reading the results gave me the impression that there might have been a translational gap between the existence of bereavement support services and its actual use. I would recommend, to broaden up the discussion about this a bit more. Is there anything known (from your or from other studies) about possible barriers for support service uptake (e.g. lack of knowldege about bereavement support programs, anxiety of being stigmatized…). What makes it especially difficult for the young to take up support services? What could be done to your opinion to overcome a possible translational gap (e.g. involvement of teachers, family doctors)?

Response: Thank you – we have widened up this discussion as follows: 

While there are many support services available [210], often the participants were either not offered support, or declined to take it up for fear of stigma, or that they would not be able to cope or did not feel it necessary. Several mentioned being handed a leaflet or book-let, which they felt was not sufficient. Barriers to accessing bereavement services have been found elsewhere and include the availability of services, a lack of knowledge of services available, unsure on how to access services, and not feeling comfortable in asking for support [43]. Future research should investigate this translational gap, and the interplay between support received from professional services, peers and the school environment [43].

  • Reviewer: In line with the remark above I would strongly recommend to rephrase the sentence in the abstract “Talking therapies are seen as insufficient.“ From reading the quotes I believe that this is not exactly true, because study participants were not attending services and then evaluating them as being insufficient. It is my impression, that the quotes rather point to the fact that talking therapies were rarely attended revealing a translational gap between existing support services and uptake.

Response: We agree with the reviewer and have amended accordingly.

  • Reviewer: The authors point to the need of alternative approaches, such as music or physical activity. This is an important finding. Are there any data about the use of E-Mental health approaches for bereavement in this context (e.g. online self-help, or internet-based CBT)? Especially young people might be attracted to these approaches. 

Response: These are important considerations and we have elaborated this section to also include e-support and expressive writing.

Reviewer 2 Report

I believe that the topic of the paper is relevant and interesting. The design of the study is appropriate and carefully conducted.  The manuscript is written clearly.

However it would be useful if authors could clarify are those different types of parental bereavements represented specific experiences of different participants or the same participant can have different experiences in different times or stages of bereavement?  If participants can have different combinations of those types of reactions, what helps them to move from one to another experience? Which factors contribute to different types or different combinations of reactions?

Obviously, there are different approaches to psychological interventions. What makes them specific to the specific experience?

Speaking about the right time for intervention, it would be useful to discuss how to assess when is the right time:  when to wait for a client to be ready for it;  when he/she needed help but doesn’t want it.

Author Response

Reviewer: I believe that the topic of the paper is relevant and interesting. The design of the study is appropriate and carefully conducted.  The manuscript is written clearly.

Response: We thank the review for their positive comments.

Reviewer: However it would be useful if authors could clarify are those different types of parental bereavements represented specific experiences of different participants or the same participant can have different experiences in different times or stages of bereavement?  If participants can have different combinations of those types of reactions, what helps them to move from one to another experience? Which factors contribute to different types or different combinations of reactions? Obviously, there are different approaches to psychological interventions. What makes them specific to the specific experience?

Response: Thank you for these helpful thoughts for consideration. The experience of bereavement is unique to each person, and there is no known one defining experience or duration, which we have made clearer in the introduction’. This research aimed to look at the experience as a whole, as a reflection, at least five years after it had occurred, in an attempt to offer insight to those who develop interventions in this area.

Reviewer: Speaking about the right time for intervention, it would be useful to discuss how to assess when is the right time:  when to wait for a client to be ready for it;  when he/she needed help but doesn’t want it.

Response: This is an important research question, and goes beyond the scope of this research. However, we have added this as a point of discussion and future research.

Reviewer 3 Report

My first observation concerns the inclusion criteria (point 2.2), which stipulate that participants have experienced mourning between the ages of 10 and 24. I believe it is a very mixed sample, since the impact of bereavement on a 10-year-old child is different from that on a teenager or young person, and because it is also influenced by the level of development, developmental as well as emotional needs of the person that the role that the dead parent had with him undergoes him. Therefore, I invite you to explain both the reasons for having considered such a vast period of time, and if and how you compare the data with each other. Here are some bibliographical references that I have consulted:

  • J. Cerel et al., “Childhood bereavement Psychopathology in the 2 years postparental death”, Journal of the American Academy of Child and Adolescent Psychiatry, vol. 45, June 2006, p. 681-690.

  • J.B. Kaplow et al., "DSM-V Diagnostic Criteria for Bereavement-Related Disorders in Children and Adolescents: Developmental Considerations", Psychiatry Interpersonal & Biological Processes, 75 (3), 2012

  • Nadine M. Melhem et al., “Grief in Children and Adolescents Bereaved by Sudden Parental Death”, Arch. Gen. Psychiatry, 68 (9), Sep. 2011, p. 911-919

  • T. Otowa et al., “The impact of childhood parental loss on risk for mood, anxiety and substance use disorders in a population based sample of male twins”, Psychiatry Research, December 15th 2014, p. 404-409.

Regarding your choice to consider participants in the study those who have experienced the death of a parent at least five years before recruitment, I suggest you indicate whether the choice of 5 years has a basis in scientific literature and which one.

Furthermore, from the data in table 1, it is observed that the years since death range from 7 to 18, a vast time span. I think it may be helpful to explain:

  • if you have considered the impact of the passage of time (and therefore of opportunities for change) on the experience of grief;

  • the impact of protective and risk factors on the bereavement experience;

  • if you have considered the possibility that the experience at 7 years after bereavement is different from that at 18 after bereavement.

Potentially, it would have been useful to note:

  • protective factors (e.g. the presence of family social support, good communication with the surviving parent)

  • and risk factors (e.g. socio-economic difficulties, experiences of illness of other family members, depression of the surviving parent);

both capable of influencing the experience of mourning, that is, favoring either its elaboration or, on the contrary, the psychopathological outcomes, as the literature on the subject indicates. All this would be useful for better interpreting the data.

The seven themes that are highlighted by the study are referred by the authors as a whole to the experience of mourning. On the other hand, it may be useful to distinguish between normal and complicated grief, contemplated in the DSM-V.

From the “results” paragraph to the end, there is a long list of the answers the participants gave to the interviews. These responses have a great emotional impact but for the purpose of publication a further step should be considered, that is, a systematization of the data, in the form of a table or diagram, for example.

Author Response

Reviewer: My first observation concerns the inclusion criteria (point 2.2), which stipulate that participants have experienced mourning between the ages of 10 and 24. I believe it is a very mixed sample, since the impact of bereavement on a 10-year-old child is different from that on a teenager or young person, and because it is also influenced by the level of development, developmental as well as emotional needs of the person that the role that the dead parent had with him undergoes him. Therefore, I invite you to explain both the reasons for having considered such a vast period of time, and if and how you compare the data with each other. Here are some bibliographical references that I have consulted: J. Cerel et al., “Childhood bereavement Psychopathology in the 2 years postparental death”, Journal of the American Academy of Child and Adolescent Psychiatry, vol. 45, June 2006, p. 681-690.; J.B. Kaplow et al., " DSM-V Diagnostic Criteria for Bereavement-Related Disorders in Children and Adolescents: Developmental Considerations ", Psychiatry Interpersonal & Biological Processes, 75 (3), 2012; Nadine M. Melhem et al., “Grief in Children and Adolescents Bereaved by Sudden Parental Death”, Arch. Gen. Psychiatry, 68 (9), Sep. 2011, p. 911-919; T. Otowa et al., “The impact of childhood parental loss on risk for mood, anxiety and substance use disorders in a population based sample of male twins”, Psychiatry Research, December 15th 2014, p. 404-409.

Regarding your choice to consider participants in the study those who have experienced the death of a parent at least five years before recruitment, I suggest you indicate whether the choice of 5 years has a basis in scientific literature and which one. Furthermore, from the data in table 1, it is observed that the years since death range from 7 to 18, a vast time span. I think it may be helpful to explain: if you have considered the impact of the passage of time (and therefore of opportunities for change) on the experience of grief; the impact of protective and risk factors on the bereavement experience; if you have considered the possibility that the experience at 7 years after bereavement is different from that at 18 after bereavement.  Potentially, it would have been useful to note: protective factors (e.g. the presence of family social support, good communication with the surviving parent); and risk factors (e.g. socio-economic difficulties, experiences of illness of other family members, depression of the surviving parent); both capable of influencing the experience of mourning, that is, favoring either its elaboration or, on the contrary, the psychopathological outcomes, as the literature on the subject indicates. All this would be useful for better interpreting the data.

Response: We thank the reviewer for these helpful reflections and opening a discussion on our inclusion criteria. In the UK, the commissioning for Children and Young People’s (CYP) services is combined, and often, interventions that are delivered for bereavement have a wide age range, and access point regarding the time since death (see Bergman, A. S., Axberg, U., & Hanson, E. (2017). When a parent dies–a systematic review of the effects of support programs for parentally bereaved children and their caregivers. BMC Palliative Care, 16(1), 1-15.). Our research, therefore, wanted to understand the experience across the ages, and time since death, of young people who might present to such services to look for commonalities for service developers to consider. While we appreciate that there is merit in comparing experiences of the loss at similar ages and time points, the approach and sample size would not allow, and this would move away from our aim, to produce commonalities across these factors.  

We have made it clearer in the manuscript our justification for this age range and criteria on time since death.  While there is a dearth of research in this area, a relevant study interviewed young people who were aged between 9-25 years attending a ‘grief camp’ together, who had experienced parental death within 2-15 years. This service, and others like it, are offered to a wide range of ages, and it is therefore important to understand parental bereavement across the age range of a young person, to look for commonalities of the experience to develop support services. The World Health Organisation defines a young person between the ages of 10-24 years. We have used this age range, which is in line with similar previous research.  

Reviewer: The seven themes that are highlighted by the study are referred by the authors as a whole to the experience of mourning. On the other hand, it may be useful to distinguish between normal and complicated grief, contemplated in the DSM-V.

Response: It is beyond the scope of the research to distinguish between ‘normal’ and complicated grief/pathology. This was a retrospective study to understand experience, not to offer a diagnosis. We have however, acknowledged this valued point in the discussion.

Reviewer: From the “results” paragraph to the end, there is a long list of the answers the participants gave to the interviews. These responses have a great emotional impact but for the purpose of publication a further step should be considered, that is, a systematization of the data, in the form of a table or diagram, for example.

Response: We agree that a visual representation of the responses is valuable, and can be found in Figure 1.

Round 2

Reviewer 1 Report

I would like to thank the authors for their careful revision and have no further comments.

Reviewer 3 Report

I am pleased that the authors have accepted some suggestions, I think the article is clearer and more understandable. I wish you good work.